# Slow sink rate in floated-demersal longline and implications for seabird bycatch risk

**Yann Rouxel** [1☺¤a]*, **Rory Crawford** [1‡¤a], **Juan Pablo Forti Buratti** [2‡], **Ian R. Cleasby** [3☺¤b]

**1** BirdLife International Marine Programme, The Royal Society for the Protection of Birds Scotland, Glasgow, United Kingdom, **2** Sea Mammal Research Unit, University of St Andrews, School of Biology, Scottish Oceans Institute, St Andrews, United Kingdom, **3** RSPB Centre for Conservation Science, The Royal Society for the Protection of Birds, Sandy, United Kingdom

☺ These authors contributed equally to this work.
¤a Current address: RSPB Glasgow office, Glasgow, United Kingdom
¤b Current address: RSPB the Lodge, Sandy, United Kingdom
‡ These authors also contributed equally to this work
* yann.rouxel@rspb.org.uk

**Data Availability Statement:** We uploaded the data to the Open Science Framework (OSF). Reference and doi is below. Rouxel, Y., Crawford, R., Buratti, J. P. F., Cleasby, I. R. (2022, February

## Abstract

Bycatch of birds in longline fisheries is a global conservation issue, with between 160,000–320,000 seabirds killed each year, primarily through being caught and drowned as they attempt to snatch baits off hooks as they are set. This conservation issue has received significant recognition in southern hemisphere longline fisheries over the past several decades, largely due to the impact on highly charismatic and highly threatened birds, notably Albatrosses. As a result, the use of effective mitigation measures has been subject to fisheries regulations to reduce seabird bycatch from longliners in a number of national jurisdictions and in several Regional Fisheries Management Organisations (RMFOs). While mitigation measures have been mandated in a number of north Pacific longline fisheries, this is largely not the case in north Atlantic longline fisheries. This includes vessels using floated-demersal longlines in the North-East Atlantic longline fishery targeting European Hake *Merluccius merluccius*, in which high levels of seabird bycatch are estimated. In this paper, we analysed the sinking speed of a floated-demersal longline used to target European Hake in the off-shore waters of Scotland, to determine potential bycatch risks to seabirds. We deployed Time Depth Recorder devices at different points of the gear. We assessed how this gear performed in comparison to the best practice minimum sink rate of 0.3 m/s recommended by the Agreement on the Conservation of Albatrosses and Petrels (ACAP) to limit bird access to baited hooks. We found that the average sinking speed of the floated-demersal longline was substantially slower than the ACAP recommendation, between two and nine times slower in non-weighted parts of the gear down to 10m water depth. Our work also found that the sink rate is particularly slow in the top 2m of the water column, increasing with depth and stabilizing at depths over 10m, presumably a consequence of propeller wash behind the vessel. We calculated that the distance astern of the vessel for hooks to sink beyond susceptible seabirds' reach largely exceeds optimum coverage of best practice design Bird Scaring Lines (100 m). Our results indicate that hooks from floated-demersal longlines are therefore readily open to seabird attacks, and as a result, present a clear bycatch risk. Research is needed to adapt existing mitigation measures to floated-longlines

4). Sink rate in floated-demersal longlines. DOI 10.17605/OSF.IO/N5AXW

**Funding:** YR - Funding was received from the Centre for Environment, Fisheries and Aquaculture Science (CEFAS) UK Seafood Innovation Fund, on the Feasibility study "Developing a floated demersal longline design that minimises seabird bycatch (FS031)" [https://www.seafoodinnovation.fund/]. Funding allowed the acquisition of the Time Depth Recorder devices, their at-sea deployment on an commercial longliner and the data collection over a series of fishing operations. None of the funders had any role in study design, collection, analysis, and interpretation of data, in the writing of the report or in the decision to submit this manuscript to publication.

**Competing interests:** The authors have declared that no competing interests exist

and to develop novel mitigation approaches to improve the sink rate of the gear without impacting target fish catch.

# Introduction

Seabirds are one of the most threatened groups of birds [1] with bycatch in fisheries recognized as one of the major threats to their conservation [2]. Worldwide, longline fisheries are estimated to cause mortality of 160,000–320,000 seabirds annually, with birds primarily caught and drowned as they attempt to steal baited hooks as the line is set [3]. This global estimate includes 56,000 from the North-Eastern Atlantic Spanish demersal longline fishery alone, primarily Great Shearwater *Ardenna gravis*, though this historical figure is based on limited data and has since been reviewed down (Pep Arcos 2020, personal communication). Primarily focussed on an area of the Celtic Seas to the southwest of Ireland known as the 'Gran Sol', this fleet targets European Hake *Merluccius merluccius* and other demersal fish such as Common Ling *Molva molva* and Atlantic Cod *Gadus morhua*. Alongside Spanish-flagged vessels, this fishery also includes–for the most part–vessels flagged to France and the United Kingdom, together representing over 85% of all European Hake landings in 2018 for ICES area 27 [4]. Recent publication estimates that up to 10,000 seabirds, including 2,600–9,000 Northern Fulmars *Fulmarus glacialis* [5], could be bycaught each year by the British flagged segment of the fleet, which represents only about 13% of the fleet's Hake landings [4]. The total magnitude of seabird bycatch across this multi-national fleet is unknown but is likely to result in the mortality of tens of thousands of seabirds each year [3].

Several simple and effective seabird bycatch mitigation measures for longline fisheries already exist [6] but most have been developed and tested in the southern hemisphere longline fisheries [7–10]. To date, limited work has been conducted to test, adapt or implement these measures for the unique set of challenges posed by 'floated' demersal longlines (also referred to as '*Spanish longline*' [11], '*piedra bola*' [12] or *'semi-pelagic longline'* [13]), which are predominantly used in the North-East Atlantic to target Hake and other whitefish fisheries. This fishery operates mainly in the ecoregion known as the Celtic Seas (ranging from the north of the Shetland Islands to Brittany in the south [14]) and towards the Bay of Biscay, including the area referred to as the '*Gran Sol*' [3,15]. 'Floated' demersal longlines are lofted off the seabed through the addition of subsurface floats on the main line (Fig 1). Due to the added buoyancy, this gear presents an elevated risk to seabirds because the floats counteract the weights attached to the line, resulting in hooks sinking slowly and making baits available to seabirds' attacks for longer periods of time. One study reported birds attacking baited hooks on floated longlines ten times more compared to longlines without floats [16].

In terms of regulation, use of bird-scaring lines have been mandated for Alaskan longline fleets since 2004, which helped reduce bycatch of surface-foraging seabirds by 88–100% in the Sablefish and Pacific cod fisheries [17,18]. In Europe, the European Union (EU) published in 2012 an Action Plan for reducing bycatch of seabirds in fishing gears, notably requiring that member states implement at least two proven mitigation measures in longline fisheries–specifically mentioning *"the Gran Sol [. . .] and non-EU waters"*–such as Night setting, Bird-scaring lines or Line weighting in accordance with minimum technical standards as set out in ACAP guidelines [19]. In spite of this action plan, Mitchell et al. [20] found that those aforementioned mitigation measures are still not required to be implemented in most EU longline fisheries and "*no action has been taken to address seabird bycatch in the Gran Sol fishery*". The 2019 EU

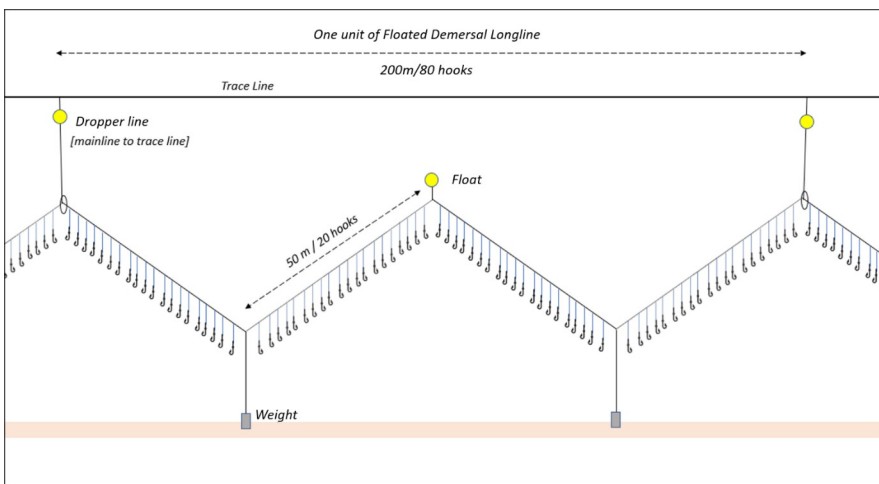

**Fig 1. Schematic of floated-demersal longline.** This 'typical' configuration is used to target European Hake in the North-East Atlantic fishery, composed by a sequence of floats, weights and dropper lines (connecting the main line–with hooks–to the trace line). A "unit" (or box) corresponds to 200m of line and 80 hooks, and a longline is comprised of up to 130 units in this fishery. Noting variation in some of the specific elements depending on fishing conditions and skipper preference.

*Technical Measures Regulation* should have provided an additional impetus for efforts to address bycatch, yet actions are still severely lacking [20]. Since the withdrawal of the UK from the EU in 2020, no concrete actions have been mandated or encouraged to tackle seabird bycatch in UK waters, by national or foreign fleets. Although a UK Seabird Bycatch Plan of Action has been in development over the past couple of years, and collaborative research platforms recently put in place (*Clean Catch UK*) [21], it remains the case that there is no clear pathway or timeline for implementation of best practice mitigation measures in the longline fleets operating in UK waters (personal communication with Defra representatives, 2021). Some vessels in the UK fleet are voluntarily using night setting and bird-scaring lines to reduce seabird bycatch [5], though the efficacy of these lines is currently unknown when paired with floated demersal longlines.

In this study, we used time-depth recorders (TDRs) to better understand the sink rate of the typical gear used by the north east Atlantic floated demersal longline fleet to inform relevant actions to reduce seabird bycatch. The Agreement on the Conservation of Albatrosses and Petrels (ACAP) defines the best practice longline sink rate as at least 0.3 meters per second to avoid seabird captures [22,23]. This can be achieved through a range of weighting options, but 5 to 8 kg weights spaced every 40 m is a configuration that is mandated for fisheries operating in the Convention on the Conservation of Antarctic Marine Living Resources (CCAMLR) area [24]. Weights that were used in the fishery in the present study can vary but, generally speaking, are substantially lighter and spaced at wider intervals; 3kg weights spaced every 100m appears as a common configuration in this fishery (Fig 1) (Juan Pablo Forti Buratti, fishery observer, personal observations 2020). With the extra buoyancy created by the regular deployment of floats along the line, it is expected that the sink rate would be substantially slower compared to best practice recommendations.

Bird behaviour and diving proficiency determines a "safe depth", below which seabird bycatch risk is greatly reduced [25]. Northern Fulmars, being surface feeders with limited diving capacity [26], are most at risk of bycatch in the upper 2 metres of the water column [27]. Other species such as Great Shearwaters can dive down to nearly 20m, although about half of dives occur within the first two meters of water, with nearly all remaining dives within 2–10 m

[28]. In this fishery, most bycatch is therefore likely occurring within the top 10m of the water column, and is therefore the key area of interest when examining the sink rate of the gear.

## Materials and methods

### Ethics statement

This research did not involve animal or human subjects, and therefore did not require ethical approval. The research was conducted in accordance with the Centre for Environment, Fisheries and Aquaculture Science (Cefas) Seafood Innovation Fund (SIF) guidelines for the project FS031, as well as the Royal Society for the Protection of Birds (RSPB) Ethics Advisory Committee guidelines

### Field work methodology

To measure the sink rate of a floated-demersal longline, we collaborated with a 40 m long commercial longliner targeting European Hake in the North-East Atlantic waters. Data collection took place between February 24[th] and March 5[th] 2020, at Latitudes 60.1442/60.2876 and Longitudes -4.4957/-4.0434 (Fig 2). All data were collected by a single observer and onboard the same vessel. We deployed a series of G5 Long-life 8MB CEFAS Time Depth Recorders (TDRs), at four different positions across the fishing gear: Dropper, Float, Middle and Weight (Fig 3). Configuration of the fishing gear, such as the spacing of weights and floats, is subject to change depending on fishing conditions and the skipper's experience, and as such there is no 'standard' configuration of a floated-demersal longline in this fleet. However, these trials were conducted on the most common configuration of the gear according to the fishery observer, and echoes the configuration recorded in other semi-pelagic Hake fisheries across the Atlantic [29–31].

Fishers were asked to deploy gear as they would in a normal fishing operation. Eight to ten TDRs were deployed on each experimental set, depending on fishing conditions. After each set was hauled, TDR data were downloaded using the DST Host software from CEFAS (https://www.cefastechnology.co.uk/downloads), then recalibrated for the next fishing operation. Each TDR was calibrated with the *Dive Logging* option, using wet/dry sensors (activation by entry into the water) and 2Hz log rate (recording every 0.5 sec), as well as 12-bit Data Points defined resolution for the Pressure sensor. TDRs were calibrated with 50 bar sensors, providing a pressure resolution better than 15cm with the above setting.

We analysed the data according to the "bycatch risk depths" for Northern Fulmar and Great Shearwater, which have constituted over 90% of all bycaught birds in published studies of this fishery [3,5], though it should be noted that overall this is a very data-poor fishery with regard to seabird bycatch. Other seabird species are being bycaught, though these are also primarily surface-feeding and pursuit-plunging species yet capable of deep diving [27,28]. We therefore split the water column into different depth sections to better understand the sinking dynamics of the gear in relation to the most affected seabird's feeding strategies [32]. Those sections are 0 to 2m (to represent surface-feeders) and 0 to 10m (for pursuit-plunging divers) [33]. We also decided to include a 0 to 5m section to provide a more precise understanding of the bycatch risk across the water column. Analysis at 2 to 5m and 5 to 10m, will also help understand the sinking dynamic of the line at different depth sections and inform potential re-design of the gear.

### Statistical analysis

**Sink rate and distance from vessel.** Sink rates were calculated as the time taken in seconds for TDRs set at different positions on the longline to traverse a specific water depth range

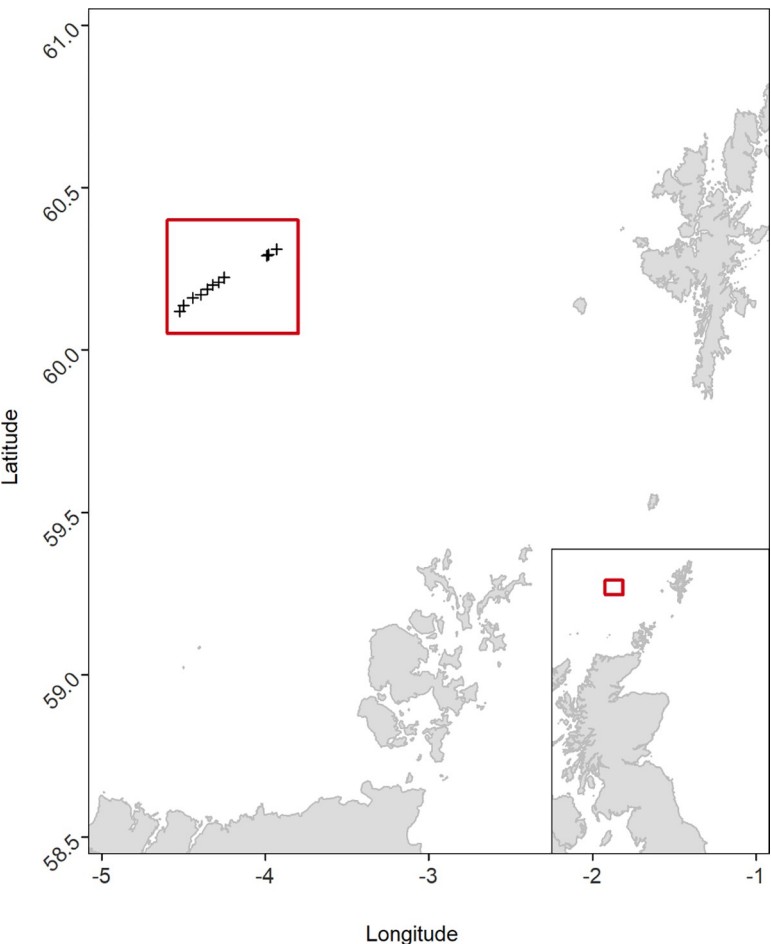

**Fig 2. Location of data collection (red square), which took place on a longliner targeting European Hake in Scottish waters, between February and March 2020.** Each black cross indicates the start of a longline deployment with TDRs attached.

(0–2 metres, 0–5 metres, 0–10 metres, 2–5 metres and 5–10 metres); separate models were run for each water depth range. The time taken to traverse each specified water depth range (sink time) was log-transformed prior to analysis and modelled using a Bayesian random effects models in the R Environment (R Version 4.0.4 [34]) using the MCMCglmm package [35].

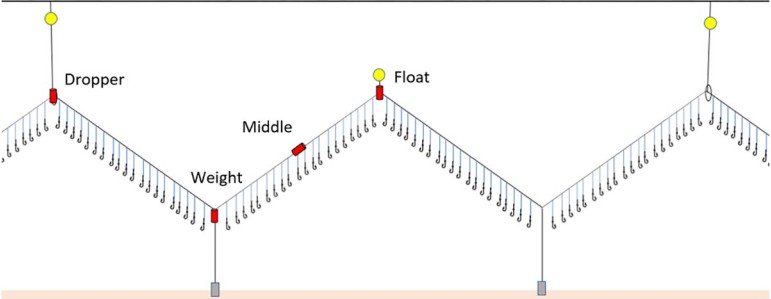

**Fig 3. Time Depth Recorders' deployment on the fishing gear.** The red cylinders indicate where the Time Depth Recorders were deployed on the floated demersal longline.

Sink time was log-transformed to prevent models predicting negative values which are unrealistic in this context. When reporting the results from Bayesian analysis we report parameter estimates (posterior means) and corresponding equal-tail 95% credibility interval (CRI) values.

As predictor variables in our initial sink rate models, we included the position of the TDR on the line (Dropper, Float, Middle, Weight) as well as the wind speed recorded during setting (wind measured on the Beaufort scale), setting speed, water depth and the log number of hooks on the line. We chose to model the number of hooks on the log scale as the raw number of hooks per net was typically very high with some discontinuity between values (range: 4000–10400) and we felt results would therefore be more interpretable on the log-scale. For ease of interpretation, setting speed and water depth were standardized (mean centred and divided by their standard deviation) prior to modelling and wind speed was centred to the modal wind speed recorded [36]. Because we were specifically interested in calculating sink rates at different positions of the gear, we did not consider the position term for removal when performing model selection. However, we decided to retain or remove the other predictors examined using leave-one-out cross-validation (LOO CV) and assessing whether their addition improved predictive performance versus a model in which only TDR position was included as a predictor. TDRs set on the same line on the same day are unlikely to be completely independent, therefore we included a random intercept for Day in our models. As one longline was set per day during the study (Table 1) this ensures measurements from all TDRs from the same deployment are grouped together to avoid potential pseudo-replication issues. Initial visual inspection of raw time to depth data suggested that the variation in sink time was greater at some positions of the gear than others. Therefore, we allowed the residual variance in our models to vary in relation to the position of the TDR to avoid potential problems with heteroscedasticity [37,38]. As before, LOO CV was used to examine whether a model in which the residual variance for each TDR position was modelled separately performed better than one in which a single, pooled estimate for the residual variation was calculated. Because time to depth was modelled as log-transformed it was back-transformed to the original scale using standard formulae [39,40]. Here we report marginal estimates from our models by performing back-transformation over the random effects distribution in our models. Marginal estimates of time to depth were also converted to estimates of sink rates (m / s). Because we are using Bayesian

**Table 1. Summary of TDRs deployment with gear configuration and fishing conditions.**

| Day | Date | Start Fishing Operation | Shooting speed [knots] | Water depth [m] | Beaufort | # hooks set | # working TDRs recording |
|---|---|---|---|---|---|---|---|
| 1 | 24/02/2020 | 03:00:00 | 7.6 | 277 | 2 | 9600 | 9* |
| 2 | 25/02/2020 | 03:15:00 | 7.7 | 271 | 2 | 9600 | 8 |
| 3 | 26/02/2020 | 04:40:00 | 7.8 | 250 | 3 | 2000 | 7** |
| 4 | 27/02/2020 | 04:00:00 | 9.1 | 255 | 4 | 2800 | 8 |
| 5 | 28/02/2020 | 04:40:00 | 8.2 | 262 | 3 | 5600 | 7** |
| 6 | 29/02/2020 | 03:00:00 | 8.5 | 265 | 7 | 10400 | 8 |
| 7 | 01/03/2020 | 04:30:00 | 8.4 | 275 | 6 | 4000 | 7** |
| 8 | 02/03/2020 | 04:20:00 | 8.4 | 270 | 4 | 4000 | 8 |
| 9 | 03/03/2020 | 04:10:00 | 8.5 | 270 | 2 | 4000 | 10 |
| 10 | 04/03/2020 | 03:20:00 | 6.5 | 260 | 2 | 10400 | 9* |
| 11 | 05/03/2020 | 03:10:00 | 6.4 | 270 | 2 | 10000 | 10 |

*a TDR was lost during the deployment.

**a TDR failed at recording.

models sink rates can be calculated across model posterior distributions allowing easy calculation of 95% credible intervals (95% CRI).

A similar approach was used to model the expected distance from the stern that different positions within the longline would attain after sinking to a specific water depth (2m, 5m, 10m). Distance from the stern in metres was estimated from raw TDR data by multiplying the number of seconds it took each TDR to reach a specified water depth by the shoot speed (m / s). Distance from stern was modelled as a log-transformed variable using the same fixed and random effects structures and modelling procedure as described above.

For all the fixed effect predictors in our models we used normal priors with a mean of 0 and a variance of 100. Priors for variance components were set using an inverse-Wishart distribution. For a single variance component, the inverse-Wishart distribution was dictated by two parameters, $V$ and $v$, in MCMCglmm notation following Hadfield [35]. Hadfield refers to $v$ as the 'degree of belief parameter' and smaller values of $v$ denote weaker belief in prior values for the unknown variance parameter $V$. We set $v = 0.002$ to create a diffuse prior for unknown variance components and ran 3 MCMC chains that began at dispersed values for $V$ (0.05, 0.5, and 1.0 respectively, with $v$ fixed at 0.002). Convergence of chains was examined through inspection of trace plots and calculation of the Gelman-Rubin statistic [41]. Chains were run for 20000 rounds, with a burn-in of 3000 rounds and a thinning interval of 10 to ensure each parameter had an effective sample between 1000–2000 per MCMC chain.

**Change in depth over time.** In addition to investigating the time it took to reach specific water depths, we modelled TDR depth over time in a continuous manner to highlight potential non-linearities in sink rate. A generalized additive mixed model (GAMM) was used with log-transformed TDR depth recorded at regular one second intervals as the response variable [42,43]. As before, log transformation was used to prevent models predicting negative values for depth and model predictions were subsequently back-transformed to the data scale. We used time elapsed since shot (seconds) as a predictor variable. The relationship between log-depth and time elapsed since shot was modelled using a spline smoother (cubic regression spine with shrinkage). We used estimated separate smoothers for each TDR position category and included a random intercept for Day. Temporal autocorrelation in depth over time was modelled by including an auto-correlation structure of order 1 (AR-1 [44]) for measures taken from each unique TDR deployment (a combination of TDR position and day of deployment). The extent of auto-correlation was then estimated separately across each TDR position category.

## Results

TDRs were deployed 97 times on the gear across 11 fishing operations. Each fishing operation consisted of a full longline deployment, ranging from 2,800 to 10,400 hooks per operation. Between seven and ten TDR recordings were collected for each fishing operation (Table 1). Two TDRs were lost across the project duration (due to fishing gear breakage) and a further four recordings failed due to errors in setting the recording periods. In total, 91 TDR deployments provided recordings (~94% deployment success) of sink rate at the four different gear positions: Dropper n = 21, Float n = 21, Middle n = 6 and Weight n = 43.

## Sink rate

With the exception of wind speed, none of the predictor variables we initially included alongside TDR position were retained in our final models after our model selection procedure (see model selection tables in the Supporting Information). Wind speed was found to be positively associated with the time taken to sink from 0–5 metres (S4 Table), from 0–10 metres (S6 Table)

**Table 2. Summary table of time taken in seconds for TDRs set at different longline locations to reach specific water depths.**

|  | Dropper | Float | Middle | Weight |
|---|---|---|---|---|
| 2m | 40.1 | 58.8 | 28.07 | 7.55 |
|  | (30.96–51.71) | (46.33–72.71) | (20.67–42.81) | (5.87–9.49) |
| 5m | 59.12 | 74.43 | 41.56 | 11.86 |
|  | (49.58–70.33) | (61.84–89.55) | (30.48–61.14) | (9.81–14.29) |
| 10m | 107.76 | 101.65 | 61.14 | 22.17 |
|  | (80.53–142.22) | (76.93–136.06) | (45.23–86.09) | (16.52–29.27) |

Values correspond to modelled estimates of average taken across all deployed TDR with the relevant longline location category. Bayesian 95% CRI are also displayed in brackets. For more detailed tables see S2, S4 and S6 Tables in Supporting Information.

and from 2–5 metres (S8 Table). Model coefficients suggest a one-unit increase on the Beaufort scale is predicted to lead to a small decrease in sink rate of ~ 3–5% in these models. Wind speed was also found to be positively associated with the expected distance travelled from stern before TDRs reached 2 metres (S14 Table), 5 metres (S16 Table), and 10 metres of depth (S18 Table). Model coefficients suggest a one-unit increase on the Beaufort scale is predicted to lead to an increase in the distance travelled before reaching these depths as ~ 6–8% in these models.

TDRs at the weight reached 2 m, 5 m and 10 m much faster than those at other positions (Table 2). Average sinking speed was significantly slower than the ACAP minimum threshold of 0.3 m/s for most parts of the gear and at any depth (Table 3, Fig 4). Only TDRs closer to the weights achieved >0.3m/s sinking speed (with the likely exception at depth 0–2 m), while TDRs set at the dropper, float or middle positions, were much slower: achieving, respectively, only 16%, 11% and 24% (on average) of the recommended sinking speed within the first two meters of the water column, 50%, 60% and 73% respectively in the 2 to 5m water depth section, and 32%, 63% and 83% in the 5 to 10m water depth section (Table 3).

Based on the outputs of our GAMM modelling Fig 5 shows the typical sinking profile over time of TDRs deployed at each longline location (see S11 Table for detailed output from GAMM modelling). The average sink rate at the dropper and middle positions never exceeded 0.3 m / s. Typically, the average sink rate at the float position was under 0.3 m / s as well. Between 100 and 125 seconds after setting, the average sink rate at the float was predicted to be at 0.3 m / s or above, but only at depths already out of susceptible seabirds' reach (around 10 m and over). Additionally, this should be interpreted with caution as the 95% CI were also wider

**Table 3. Estimated sinking speed across different water depth ranges expressed as the average sink rate (m/s) at each TDR location together with 95% CRI in brackets.**

| Depth | Dropper | % of 0.3m/s | Float | % of 0.3m/s | Middle | % of 0.3m/s | Weight | % of 0.3m/s |
|---|---|---|---|---|---|---|---|---|
| 0-2m | 0.049 | 16 | 0.034 | 11 | 0.071 | 24 | 0.26 | 87 |
|  | (0.039–0.064) |  | (0.027–0.043) |  | (0.046–0.099) |  | (0.21–0.34) |  |
| 2-5m | 0.15 | 50 | 0.18 | 60 | 0.22 | 73 | 0.67 | 223 |
|  | (0.12–0.19) |  | (0.14–0.22) |  | (0.16–0.28) |  | (0.54–0.86) |  |
| 5-10m | 0.097 | 32 | 0.19 | 63 | 0.25 | 83 | 0.49 | 163 |
|  | (0.067–0.13) |  | (0.15–0.25) |  | (0.18–0.32) |  | (0.41–0.61) |  |

The associated percentage of ACAP sinking speed recommendation achieved by TDRs based upon modelled sink speeds also provided. For more detailed tables see S8 & S10 Tables in Supporting Information.

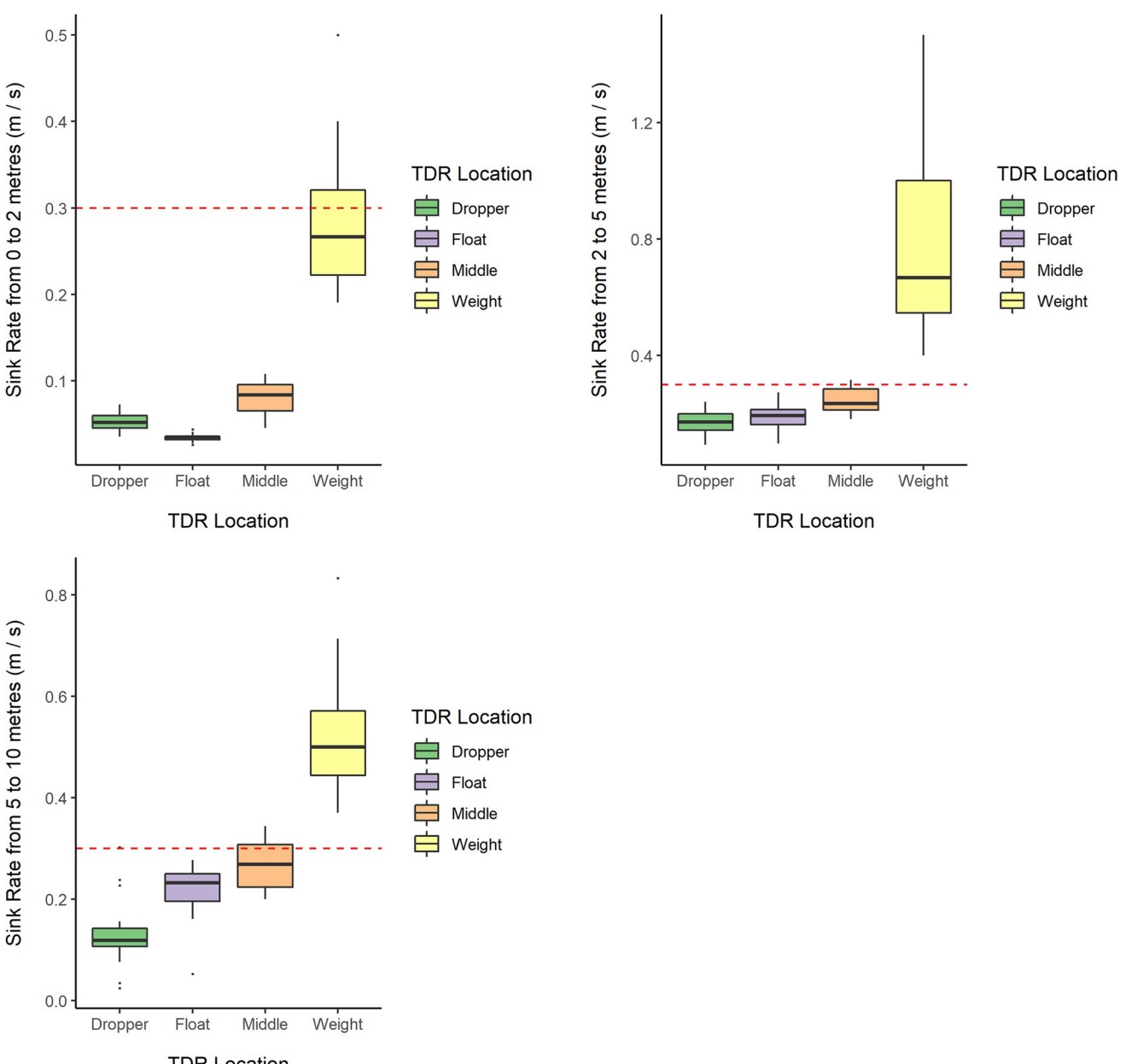

**Fig 4. Box plots showing observed sink rates (raw, unmodelled data) obtained from TDR loggers at each TDR location across specified depth ranges.** Red dotted lines indicate ACAP sinking speed recommendation (0.3 m/s).

at such depths, highlighting a greater level of uncertainty about model prediction at this point in the curve. The average sink rate at the weight location was below 0.3 m / s in the first 2 meters of the water column, then rapidly exceeded this tresholds from approximately 2 m in depth until around 13 m, then dropping again below 0. 3 m / s thereafter. However, the depth attained by a TDR at the weight location still exceeded the depth that would be attained by a longline sinking at a constant rate of 0.3 m / s during the first 50 seconds of deployment. The raw data points displayed on the plots also highlight variation between sink rates observed across different TDR deployments at the same location but performed on different days / locations.

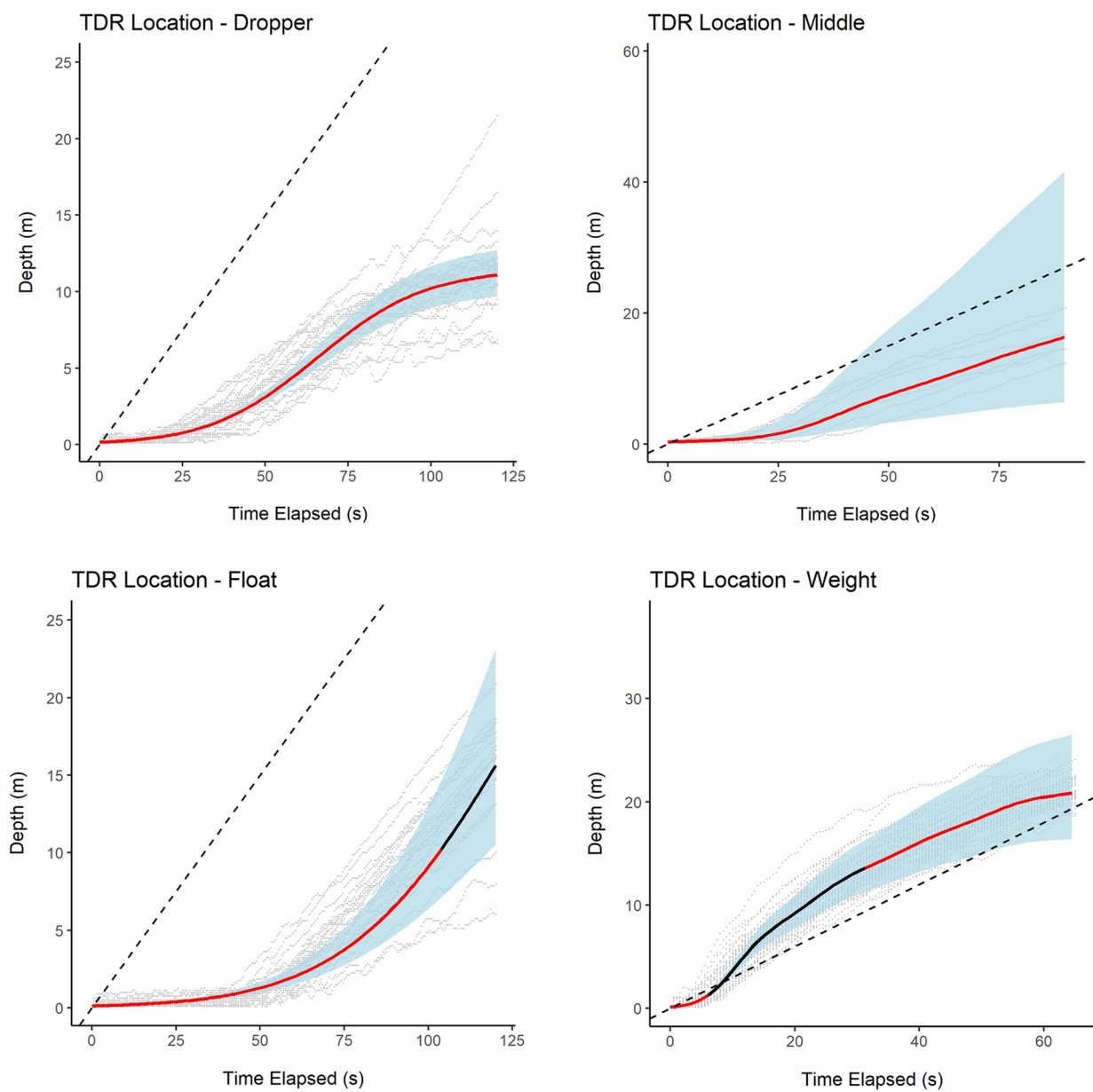

**Fig 5. The relationship between TDR depth and time elapsed since line was set at each TDR location along the longline.** The solid line represents the average relationship across all recorded TDRs deployed at a specific location estimated via a GAMM. The solid line is black when predicted sink rate is below 0.3 m/s and red when predicted sink rate is above 0.3 m / s. The blue polygon around the line encompasses the 95% confidence interval (95% CI) around model predictions. The raw data from each TDR deployment is also displayed as a grey line to highlight variation between different longline deployments on different days. The dashed diagonal line on the plot represents the trajectory that would be achieved by a hypothetical TDR sinking at a constant rate of 0.3 m / s. See S6 Table for more details.

## Distance from vessel and hooks sinking dynamics

Assuming the portions of the longline that are situated between two different sinking points of the gear (e.g. a weight and a float) follow a linear sinking trend across its length, we calculated that for the 50 m portion of the fishing line between a weight and a buoy (or dropper) which hosts 20 hooks (Fig 1), the percentage of hooks with a sink rate faster or equal to 0.3 m / s would equal 0% at 2 m depth, 20% at 5 m depth and 25% at 10m depth (S12 Table). With a

**Table 4. Summary table of expected distance travelled from vessel in metres for TDRs set at different locations to reach specific water depths.**

|  | Distance from vessel to reach 2m deep (in m) | Distance from vessel to reach 5m deep (in m) | Distance from vessel to reach 10m deep (in m) |
|---|---|---|---|
| Dropper | 158.58 | 237.87 | 443.73 |
|  | (129.21–200.63) | (188.56–304.21) | (332.91–599.04) |
| Float | 231.17 | 297.47 | 405.39 |
|  | (188.29–290.66) | (230.94–382.81) | (299.61–548.48) |
| Middle | 109.66 | 161.55 | 246.07 |
|  | (78.47–171.17) | (115.65–243.45) | (176.89–365.09) |
| Weight | 29.66 | 47.41 | 88.69 |
|  | (23.91–36.62) | (37.51–60.21) | (66.53–118.04) |

Bayesian 95% CRI are also displayed in brackets. For more detailed tables see Supporting Information.

moving vessel setting the longline at roughly 8 knots (n = 11; 7.9 knots in average during our trials (range: 6.5–9.1)), we calculated that the distance astern from the vessel for the longline to reach 2m water depth ranged from 29.66 m (weight) to 231.17 m (float), 47.41 m (weight) to 297.47 m (float) to reach 5 m, and 88.69 m (weight) to 443.73 m (dropper) to reach 10 m (Table 4). See S14, S16 & S18 Tables for more details.

## Discussion

Two characteristics of the sink profile are particularly important in assessing hook and bait accessibility for seabirds while the longline sinks within the top 10 m of water: the sink rate (or speed of descent) and the distance from the vessel's stern. The first because this relates to the diving range of the species most susceptible to capture–particularly Northern Fulmar but also Skuas, Shearwaters and Gannets–and how "easy" it is to catch a moving bait for these birds. The second because birds won't approach a vessel too closely, but mainly because current 'best practice' bird-scaring line designs only cover an area 100m astern of the vessel.

Our analysis indicates that only the hooks close to the weight-lines are likely to sink at the recommended minimum sink rate of 0.3 m / s to limit seabird bycatch. It also indicates that the sink rate slightly increases with depth, stabilizing at depths over 10m, presumably a consequence of propeller wash behind the vessel as noted in other similar studies [45]. In the surface waters (0–2 m depth), virtually all the hooks are below this sinking threshold.

We estimated that most of the longline (and therefore hooks) were, on average, between 100 m and 230 m astern of the vessel before reaching a depth of 2m, and between 150 m to 300 m, before reaching a depth of 5 m. In the most extreme cases–near the floats and Droppers–the longline may be up to 290 m astern of the vessel before reaching 2 m, 380 m before reaching 5 m, and up to 600 m before reaching a depth of 10 m. We acknowledge that such estimates are fairly coarse (multiplying setting speed by time to depth) and that additional factors might dictate how far a longline can travel horizontally. Although environmental factors were recorded during field work, we were not able to assess their influence on sinking speed. Wind speed has showed a positive relationship with hooks sinking rate in a drifting-pelagic longline fishery [46], but the drastically different weighting regime renders any assumptions hazardous in our case. Slower vessel setting speed is also known to reduce the amount of time hooks remain at the surface and at depths that seabirds can reach [47], although its effect on a floated-demersal longline and the practicality of reduced vessel speed during fishing operations needs to be investigated.

Albeit potentially incomplete, our results strongly suggest that with floated-demersal long-lines, seabirds remain at high risk of bycatch for some distance astern of the vessel. Even with a 150 m long bird scaring line–the recommended minimal length for demersal longline fisheries [48]—deployed at the correct height from the stern, only 100 m behind the vessel would be covered, leaving a large proportion of hooks available to seabirds. In spite of the reported use of bird-scaring lines by some vessels in this fleet (the exact specifications and efficacy of which are unknown in the absence of paired sea trials), the slow sinking speed of this gear–even if best practice bird-scaring lines were deployed–is likely the driver of the high seabird bycatch estimates reported by Anderson et al. [3] and Northridge et al. [5].

Another recommended mitigation measure for longline fisheries is night setting, but its effectiveness can be reduced during bright moonlight and when using powerful deck lights [49]. More importantly, the floated-demersal longline fishery frequently operates in high lati-tudes of the northeast Atlantic Ocean, including in the Faroe-Shetland channel, where during the summer months the time between nautical dusk and dawn is virtually absent—or at best—very limited. It has also been found that in the Alaskan Longline fisheries, Northern Fulmars were caught at significantly higher rates at night [50], highlighting further that best practice mitigations can vary by species assemblage and fishery.

In the absence of an easy-to-implement solution, investment in gear modification research is urgently needed to adapt existing and novel mitigation measures to this fishery. Although changes in the floated-demersal longline configuration would not be without challenges for fishers and likely be unpopular with industry. For instance, a switch from current 3kg to 5kg weights for a set of ten thousand hooks would represent roughly an additional half a tonne to be carried onboard and handled by fishers. Cortés & González-Solís [51] also found that in artisanal demersal longlines, more weight is susceptible to increase entanglement risks between the branchlines and hooks during the setting operations. However, change in the longline weighting regime is the single measure most likely to deliver a significant reduction in seabird bycatch, and a series of measures that could help reduce operational issues should be investigated such as alternative spacing between branchlines and between weights [52], use of steel weights instead of concrete [29], etc. When paired with bird scaring lines, appropriate weighting regime can virtually eliminate seabird bycatch from demersal longline fisheries [18].

Developing and testing a floated-demersal longline with a significantly improved sinking speed more in line with ACAP's recommendations, whilst maintaining or potentially improv-ing economic returns, would be a win-win scenario that is more likely to foster interest and support from the fishery. Seabirds are often able to steal numerous baits before being hooked, and the resulting loss in fish catching potential reduces the efficiency of the fishing operation [53]. Reducing bycatch could therefore bring direct economic returns, both through increased fish catch and decreased hook loss. Further, by improving the sustainability of the fishing oper-ation, the potential for these fisheries to pass third-party sustainability certification schemes is likely to increase.

## Supporting information

**S1 Table. Model selection tables for time taken to reach 2 metres in depth.** Table shows model tested and corresponding Root Mean Square Error (RMSE) calculated using Leave-One-Out Cross-Validation (LOO-CV). Best performing model highlighted in bold. $\varepsilon$ [TDR Position] denotes a model in which separate estimates of the residual variance were made for each Position.
(PNG)

**S2 Table. Time taken in seconds for TDRs set at different locations to reach 2 metres depth.** Table display coefficients from a model in which time was modelled using a log transformation. Back-transformed coefficients for sink speed also displayed in original units as well as expressed as the average sink rate (m / s) from 0–2 metre depth. σ Day is the random effect associated with day on which longlines were deployed. σε is the residual variation in the model–note that different residual variation parameters were estimated for each location in this model. n = 91 observations, 11 days. Model marginal $R^2$ = 0.93 (95% CRI: 0.89–0.95); Model conditional $R^2$ = 0.94 (0.91–0.96).
(PNG)

**S3 Table. Model selection tables for time taken to reach 5 metres in depth.** Table shows model tested and corresponding Root Mean Square Error (RMSE) calculated using Leave-One-Out Cross-Validation (LOO-CV). Best performing model highlighted in bold. ε [TDR Position] denotes a model in which separate estimates of the residual variance were made for each Position.
(PNG)

**S4 Table. Time taken in seconds for TDRs set at different locations to reach 5 metres depth.** Table display coefficients from a model in which time was modelled using a log transformation. Back-transformed coefficients for sink speed also displayed in original units as well as expressed as the average sink rate (m / s) from 0–5 metre depth. Back-transformed estimates assume Beaufort scale is set at its modal value. σ Day is the random effect associated with day on which longlines were deployed. σε is the residual variation in the model–note that different residual variation parameters were estimated for each location in this model. n = 91 observations, 11 days. Model marginal $R^2$ = 0.93 (95% CRI: 0.89–0.95); Model conditional $R^2$ = 0.94 (0.90–0.96).
(PNG)

**S5 Table. Model selection tables for time taken to reach 10 metres in depth.** Table shows model tested and corresponding Root Mean Square Error (RMSE) calculated using Leave-One-Out Cross-Validation (LOO-CV). Best performing model highlighted in bold. ε [TDR Position] denotes a model in which separate estimates of the residual variance were made for each Position.
(PNG)

**S6 Table. Time taken in seconds for TDRs set at different locations to reach 10 metres depth.** Table display coefficients from a model in which time was modelled using a log transformation. Back-transformed coefficients for sink speed also displayed in original units as well as expressed as the average sink rate (m / s) from 0–10 metre depth. Back-transformed estimates assume Beaufort scale is set at its modal value σ Day is the random effect associated with day on which longlines were deployed. σε is the residual variation in the model. n = 91 observations, 11 days. Model marginal $R^2$ = 0.92 (95% CRI: 0.88–0.94); Model conditional $R^2$ = 0.93 (0.91–0.95).
(PNG)

**S7 Table. Model selection tables for time taken to travel between 2 and 5 metres of depth.** Table shows model tested and corresponding Root Mean Square Error (RMSE) calculated using Leave-One-Out Cross-Validation (LOO-CV). Best performing model highlighted in bold. ε [TDR Position] denotes a model in which separate estimates of the residual variance were made for each Position.
(PNG)

**S8 Table. Time taken in seconds for TDRs set at different locations to travel from 2 to 5 metres depth.** Table display coefficients from a model in which time was modelled using a log transformation. Back-transformed coefficients for sink speed also displayed in original units as well as expressed as the average sink rate (m / s) from 2–5 metre depth. Back-transformed estimates assume Beaufort scale is set at its modal value. σ Day is the random effect associated with day on which nets were deployed. σε is the residual variation in the model–note that different residual variation parameters were estimated for each net location in this model. n = 91 observations, 11 days. Model marginal $R^2$ = 0.84 (95% CRI: 0.72–0.91); Model conditional $R^2$ = 0.86 (0.74–0.92).
(PNG)

**S9 Table. Model selection tables for time taken to travel between 2 and 5 metres of depth.** Table shows model tested and corresponding Root Mean Square Error (RMSE) calculated using Leave-One-Out Cross-Validation (LOO-CV). Best performing model highlighted in bold. ε [TDR Position] denotes a model in which separate estimates of the residual variance were made for each Position.
(PNG)

**S10 Table. Time taken in seconds for TDRs set at different locations to travel from 5–10 metres depth.** Table display coefficients from a model in which time was modelled using a log transformation. Back-transformed coefficients for sink speed also displayed in original units as well as expressed as the average sink rate (m / s) from 5–10 metre depth. σ Day is the random effect associated with day on which longlines were deployed. σε is the residual variation in the model–note that different residual variation parameters were estimated for each location in this model. n = 91 observations, 11 days. Model marginal $R^2$ = 0.77 (95% CRI: 0.68–0.82); Model conditional $R^2$ = 0.79 (0.69–0.83).
(PNG)

**S11 Table. Results from a GAM modelling the relationship between time since deployment and depth reached at different parts of the gear.** σ Day represents the random effect associated with longlines deployed on different days. The temporal autocorrelation in depth over time is represented as ø and was estimated separately for each gear location. Different smoothers were fitted for each gear location and details on these smoothers (estimated degrees of freedom and k index are also displayed). Model $R^2$ = 0.88.
(PNG)

**S12 Table. Calculated average sinking speed (in m/s) per hook in a 20 hooks floated-demersal longline section.** Using a linear series from TDRs recordings at different positions of the gear (in grey shading). In bold are rates equal or over ACAP recommendation (0.3 m/s).
(PNG)

**S13 Table. Model selection tables for the expected distance travelled from stern in metres for TDRs set at different positions to reach 2 metres depth.** Table shows model tested and corresponding Root Mean Square Error (RMSE) calculated using Leave-One-Out Cross-Validation (LOO-CV). Best performing model highlighted in bold. ε [TDR Position] denotes a model in which separate estimates of the residual variance were made for each Position.
(PNG)

**S14 Table. Expected distance travelled from stern in metres for TDRs set at different locations to reach 2 metres depth.** Table display coefficients from a model in which distance from stern was modelled using a log transformation. Back-transformed coefficients for distance travelled also displayed in original units. Back-transformed estimates assume Beaufort scale is

set at its modal value. σ Day is the random effect associated with day on which longlines were deployed. σε is the residual variation in the model–note that different residual variation parameters were estimated for each location in this model. n = 91 observations, 11 days. Model marginal $R^2$ = 0.93 (95% CRI: 0.91–0.95); Model conditional $R^2$ = 0.95 (0.94–0.96). (PNG)

**S15 Table. Model selection tables for the expected distance travelled from stern in metres for TDRs set at different positions to reach 5 metres depth.** Table shows model tested and corresponding Root Mean Square Error (RMSE) calculated using Leave-One-Out Cross-Validation (LOO-CV). Best performing model highlighted in bold. ε [TDR Position] denotes a model in which separate estimates of the residual variance were made for each Position. (PNG)

**S16 Table. Expected distance travelled from stern in metres for TDRs set at different locations to reach 5 metres depth.** Table display coefficients from a model in which distance from stern was modelled using a log transformation. Back-transformed coefficients for distance travelled also displayed in original units. Back-transformed estimates assume Beaufort scale is set at its model value. σ Day is the random effect associated with day on which longlines were deployed. σε is the residual variation in the model–note that different residual variation parameters were estimated for each location in this model. n = 91 observations, 11 days. Model marginal $R^2$ = 0.92 (95% CRI: 0.89–0.95); Model conditional $R^2$ = 0.95 (0.93–0.96). (PNG)

**S17 Table. Model selection tables for the expected distance travelled from stern in metres for TDRs set at different positions to reach 10 metres depth.** Table shows model tested and corresponding Root Mean Square Error (RMSE) calculated using Leave-One-Out Cross-Validation (LOO-CV). Best performing model highlighted in bold. ε [TDR Position] denotes a model in which separate estimates of the residual variance were made for each Position. (PNG)

**S18 Table. Expected distance travelled from stern in metres for TDRs set at different locations to reach 10 metres depth.** Table display coefficients from a model in which distance from stern was modelled using a log transformation. Back-transformed coefficients for distance travelled also displayed in original units. Back-transformed estimates assume Beaufort scale is set at its modal value. σ Day is the random effect associated with day on which longline were deployed. σε is the residual variation in the model. n = 91 observations, 11 days. Model marginal $R^2$ = 0.91 (95% CRI: 0.81–0.94); Model conditional $R^2$ = 0.94 (0.91–0.96). (PNG)

## Acknowledgments

We are particularly grateful for the support from the University of St. Andrews *Sea Mammal Research Unit*, which helped in securing capacity for onboard observation and data collection. Sincere thanks to Hooktone Limited, the Longliner's crew and skipper who granted us permission and technical support to operate on their vessel during real fishing conditions, without whom this project would have not been possible.

## Author Contributions

**Conceptualization:** Yann Rouxel, Rory Crawford.

**Data curation:** Ian R. Cleasby.

**Formal analysis:** Ian R. Cleasby.

**Funding acquisition:** Yann Rouxel, Rory Crawford.

**Investigation:** Juan Pablo Forti Buratti.

**Methodology:** Yann Rouxel, Juan Pablo Forti Buratti.

**Project administration:** Yann Rouxel.

**Resources:** Rory Crawford.

**Software:** Ian R. Cleasby.

**Supervision:** Yann Rouxel.

**Validation:** Yann Rouxel.

**Writing – original draft:** Yann Rouxel.

**Writing – review & editing:** Yann Rouxel, Rory Crawford, Juan Pablo Forti Buratti, Ian R. Cleasby.

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
