## [Decision Letter · Decision Letter 0]

6 Jan 2022

PONE-D-21-37923Slow sink rate in floated-demersal longlines poses high bycatch risk to seabirdsPLOS ONE

Dear Dr. Rouxel,

Thank you for submitting your manuscript to PLOS ONE. After careful consideration, we feel that it has merit but does not fully meet PLOS ONE’s publication criteria as it currently stands. Therefore, we invite you to submit a revised version of the manuscript that addresses the points raised during the review process.

We look forward to receiving your revised manuscript.

Kind regards,

Vitor Hugo Rodrigues Paiva, Ph.D.

Academic Editor

PLOS ONE

Journal Requirements:

"We are particularly grateful for the funding support from Centre for Environment, Fisheries and Aquaculture Science (CEFAS) UK Seafood Innovation Fund, which allowed us to carry out this work, as well as the University of St. Andrews Sea Mammal Research Unit, which helped in securing capacity for onboard observation and data collection. Sincere thanks to Hooktone Limited, the Longliner’s crew and skipper who granted us permission and technical support to operate on their vessel during real fishing conditions, without whom this project would have not been possible."

"YR - Funding was received from the Centre for Environment, Fisheries and Aquaculture Science (CEFAS) UK Seafood Innovation Fund, on the Feasibility study “Developing a floated demersal longline design that minimises seabird bycatch (FS031)” [https://www.seafoodinnovation.fund/]. Funding allowed the acquisition of the Time Depth Recorder devices, their at-sea deployment on an commercial longliner and the data collection over a series of fishing operations. None of the funders had any role in study design, collection, analysis, and interpretation of data, in the writing of the report or in the decision to submit this manuscript to publication."

Reviewers' comments:

Reviewer's Responses to Questions

**Comments to the Author**

1. Is the manuscript technically sound, and do the data support the conclusions?

Reviewer #1: Partly

Reviewer #2: Yes

Reviewer #3: Yes

Reviewer #4: Partly

Reviewer #5: Yes

2. Has the statistical analysis been performed appropriately and rigorously? 

Reviewer #1: No

Reviewer #2: Yes

Reviewer #3: I Don't Know

Reviewer #4: Yes

Reviewer #5: Yes

3. Have the authors made all data underlying the findings in their manuscript fully available?

Reviewer #1: No

Reviewer #2: Yes

Reviewer #3: No

Reviewer #4: Yes

Reviewer #5: Yes

4. Is the manuscript presented in an intelligible fashion and written in standard English?

Reviewer #1: Yes

Reviewer #2: Yes

Reviewer #3: Yes

Reviewer #4: Yes

Reviewer #5: Yes

5. Review Comments to the Author

Reviewer #1: In this study, the authors collected and analyzed sink rate data from a floated demersal longliner in the Eastern North Atlantic. In my opinion, this research topic is important for seabird bycatch mitigation. For example, see Dietrich et al, 2008. The study design is appropriate, although there are several places detailed below that needs to be further clarified. The statistical analysis as used in this manuscript is not satisfactory, and both the analysis and the interpretation of the results can be further improved.

In this study, the use of spline is good as it smoothes out the noise and it is also straight forward to calculate the sink rate, which is the 1st derivative of the spline. Both Fig5a and 5b look good. However, Fig5c and 5d look problematic. First, the interval estimates on the right-hand side are way too wide. Second, most data tracks stop midway. Either the depth readings didn't change afterwards or the device stopped recording at certain depth. I think the authors just chopped the second half of the data if depth readings remained stable. This is understandable, but the authors need to document it in the methods section. Assuming this is the case, the use of spline smoothing is not appropriate for the second half of the data in Fig5c and d due to this data cutting. For example, in Fig5c, to the right of ~110 on the x-axis, the spline flattens while none of the tracks flattens, and the flattening is simply due to inappropriate smoothing across multiple chopped tracks. The authors need to be careful here. Even though this portion of the data (>10m) has little practical significance to this study, both the methods and the interpretation of the result need to be correct.

In addition, what is the observed target catch and bycatch rates for these experimental trials? If the authors plan to save these statistics for another paper, at least provide a summary here.

The model selection results are missing. Add a table with the loo score for each candidate model.

Specific comments:

line 127: even though there is no standard configuration, the authors need to provide evidence that the configuration presented here is representative of the actual situation.

line 180: Where is your settings for the priors, which is an essential component of every Bayesian model?

line 181: which type of CI did you pick, symmetric or HDI?

line 203: Are these 97 trials conducted on the same fishing vessel with the help of a single observer? This is important because otherwise "observer id" and "vessel id" would be two important factors affecting the sink rate and they should be incorporated into the model.

line 262: Rephrase this sentence. I can guess what you are trying to say by looking at the table in the Appendix. Also, it is calculation not simulation.

Fig5a: Looking at this figure, I can see that either the device used in this study has a low depth resolution or the data have been truncated. State in the method section what is the design accuracy of this device and the resolution of the recorded data.

The surface layer is the most important in this study. What is the design accuracy of this device for the surface layer? Did you calibrate the device (pressure sensor) before using it or after the trials? Readings may drift under harsh environmental conditions. A seperate section on the validity of this device and its readings would further strenghen this study.

Editorial comments:

line 45: Do not capitalize the second part of a scientific name.

line 49: Use different dashes consistently

line 64: What do you mean here? "lofted off the seabird"?

line 179: not grammatically correct.

line 240: typo

Fig2: which base map did you use, where is the attribution, what projection did you use? Instead of a star, I think a rectangle shape is more appropriate. Also, it might be better to use a zoomed-in main map and keep this one as an inset.

Some references that might be useful:

Dietrich K., et al. 2008. Integrated weight longlines with paired streamer lines – Best practice to prevent seabird bycatch in demersal longline fisheries. Biological Conservation.

Reviewer #2: This is an interesting study that address an important aspect when it comes to pinpointing reasons, and consequently also effective mitigation actions, for high bycatch rates in floated-demersal longlines in the north Atlantic. I think the results are highly relevant in a broader perspective, as it gives some general estimates to the sink rate in these types of fisheries, as well as shoeing the importance of weighted lines in reducing the time period when the bait is available for seabirds. The statistics/methods seems to appropriate, and I only have minor comments. Addressing them will probably remove my small concern about the seemingly very small N for the rather complex models used. Good work.

Minor comments:

Title: Suggest to moderate the title, as it doesn’t seems like you have actually measured any changes in bycatch risk, although this is implied/assumed by the observed sink rate.

Line 51: Would analysis be a better word than “data”, in “Recent data estimate that”?

Line 62: In my printed version it seems that there is a different font or color within this line, please check

Line 120-142: It is not clear to me whether all observation is done from one vessel, or if there are multiple vessels? Suggest to include this information in the text.

Line 141-142: It is a bit difficult to understand the rationale behind the different sections chosen as basis for the different models? Especially since there seems to be no focus on this in the discussion. Why overlapping intervals, from the text it seems to me that 0-2 (surface feeders) meters and 2-10 (diving) meter would be logical choices given the two different feeding regimes described? Suggest to clarify.

Line 161: How was it standardized, and why does this standardization ease interpretation? My initial though is that it would be easier to interpret according to actual setting speed and depth rather than a (mean?)standardized version of the variables for example, as I don’t know the range of the data.

Line 166: Not sure if I understand why predictive performance is in focus in the model selection, when the results seems to be predicted over mean’s (or the range, not explained) of all other variables than the TDR-positions. Why the model selection if all variables are relevant to account for when measuring effects on sink speed? Suggest to clarify. Also, I might have overlooked it, but I cant seem to find the results from the model selection?

Line 168: would it be relevant to consider a vessel random effect as well?

Line 180: Change “used” to “using”

192: I don’t understand why a log transformation was used to prevent the model to predict negative values for depth. Or rather, I don’t understand why the model would predict negative values for depth if the proper distribution is assumed in the models. Did you assume a normal distribution, and is this assumption correct? Would a truncated distribution be a better choice? Suggest to include information about the type of model.

Line 151-200: It’s a bit unclear to me why you chose to both construct models with sink rate as a response, and models with depth over time? Why not just the latter, as it seems to provide all the information (sink rates could easily be derived from these), and why not include TDR location as an interaction effect with time elapsed rather than creating 4 different models?

Also, from the results it seems that you only have 91 lines in total (and much less per TDR location (?)), did you actually have enough statistical power to make sound estimate for all models/combinations. Your model-output seems to suggest that you have, but the model structure seems rather complex given the potential range for the different variable levels (e.g. as low as 6? for “middle”), and N in total? I have probably misunderstood this part, so please clarify in the text.

Line 202: I miss a short description of which parameters comes into play in the different models (except for TDR-position, all seems to be omitted from the text (expect from being included in the sup. tables). It is thus difficult so see the rationale for doing a model selection at all, if these variables are of no interest.

Figures: Are these the predicted response across the range of all other variables, or the for example the mean? Please clarify.

Reviewer #3: This manuscript presents an assessment of the sinking speed of floated-demersal longline to determine potential bycatch risks to seabirds. First of all, I would like to present my congratulations to the author by this interesting and hard piece of work. Generally, the manuscript is well written and easy to understand. The main claim of the manuscript is well defined and properly placed in the context of the reviewed literature. However, despite data is likely to support the claims, some more details on the results are needed, namely in terms of the modelling exercises, how final models fit each sample and which predictor variables were included. At the end it was hard to find any detail on predicted variables other than TDR position or water depth, i.e. wind speed, setting speed and number of hooks. Such lack of details prevents the assessment of the appropriate and rigorous performance of the statistical analysis. Some details on GAMM results might be available under the supplementary material which as reviewer I had no access. Please find bellow my detailed revision:

Introduction

Line 44 Consider to replace "Puffinus Gravis" by "Ardenna gravis"

Line 46 Replace "Celtic Sea" by "Celtic Seas"

Line 69 to 87 Consider to move this paragraph to a later section of your discussion. Here would be more interesting to introducing the reader to a narrow scope of your main topic/goal.

Methods

Line 139 and 140 I found this affirmation to important to be based in such weak peaces of information reflecting a notable lack of data. Remembering the Gran Sol bycatch figures are based in very low observation effort, from only one vessel. Despite, there are strong evidences of seabird bycatch to occur in the area, listing top bycatch species and bycatch numbers in the north east Atlantic floated demersal longline fishery based in such little information requires care. There is not much available information on how Great Shearwater bycatch rates were estimated for the Spanish fleet operating in Gran Sole, one important aspect to be into consideration is the presence of birds in the area during a narrow period of time. In the other hand, there are published evidences showing Northern Gannets mainly followed by Cory's Shearwater are being caught in serious numbers in longlines. Perhaps replace the assumption of "most bycaught species" by "two of the known bycaugh species" fits better the state of the art. Also, after reading the introduction and the methods I was expecting a need from the authors to discuss their findings in terms of any king of ecological or biological aspects of Fulmar or Great Shearwater. But, I understood those species are only used to justify the chosen distance bands. Perhaps, authors might be wider and justify their distance bands choice by the existence of different feeding strategies in seabird species. There are some classifications proposed by other authors, e.g. surface feeders, pelagic feeders, etc. And perhaps using Fulmar and Great Shearwater as an example of those groups, and add others if relevant.

Line 160 Please clarify the choice of using log number of hooks instead the raw number.

Line 161 Please detail the standardized method used for setting speed and water depth

Results

Line 211 Table 1 - to confirm within the journal guidelines about measurement units, knots are not an IS unit; Perhaps it is worth to remove "Species targeted", because all samples were targeting hake.

214 What about the results of the modelling? How much the predictor variables explain the sink rate variability? Which predictor variables were taking into account in the final model? Only the effect of position of TDR on the line on sink rates are given. Perhaps the results of the LOO CV step to retain or remove predictors are missing, but important to clarify the authors' choices. More details on those results are needed.

Line 237 As stating before, apart from the outputs of GAMM modelling, the results of the modelling exercise itself will help reader to understand the goodness of fit of the selected model.

Line 255 It is a very non-important detail, but as sink rate is expectable better when above 0.3m/s my suggestion is to illustrate it with a black line, and when bellow as a red line.

Line 262 Still this paragraph about sink rate or more about "Distance from stern for specific water depth ranges". Consider to add this "sub-chapter" here or follow the same approach used in the Methods - "Sink Rate and Distance from..." - for the entire sub-chapter

266 Authors use knots, please confirm guidelines for the need of using IS units

Discussion

Line 275 to 324 Depending on the results, it would be worth to discuss the possible effects of wind speed, setting speed and number of hooks (as proxy for size of a longline) on the sink rates and then on the bycatch risk.

Also, only one vessel was sampled in this study. Could such fact be a limitation for this study? Might be desired to discuss it in the light of the variability of longlines even within the same fishing fleet.

Lines 313 to 316 Could authors discuss the main challenges for fishers regarding changes in the longline weighting?

Reviewer #4: General issues:

Rouxel et al. present a method to assess the sink rate of a floated-demersal longline, and apply this method to a longline used to target European Hake in the offshore waters of Scotland, and find that the sink rate is slower than the rate recommended by the ACAP, in the top 2m of the water column in particular, and the distance astern of the vessel for hooks to sink beyond seabirds’ reach largely exceeds the optimum coverage. They indicate that hooks from floated-demersal longlines present a clear bycatch risk.

I found the experimental method adequately motivated, and I believe it will be useful in the following seabird bycatch analyses. I have some concerns about the sufficiency of an experiment on a single longline to achieve a general conclusion. Moreover, there is no observation on seabird bycatch during this experiment, and no analysis on the impacts of sink rate and other factors on seabird bycatch, so it is a hypothesis that the slow sink rate in this floated-demersal longline may pose high bycatch risk to seabirds, but to what extent, we don’t know. I imagine that this kind of fundamental experiment will be useful if a following analysis is conducted to relate seabird bycatch to the sink rate. At this point, I suggest to revise the title to avoid overstating this study and add more references in the discussion to emphasize the potential impacts of sink rate on seabird bycatch. All concerns should be addressed via major revision.

Specific issues:

Lines 41-43: Where were these numbers (160,000-320,000) from?

Line 76: Please specify the best practice mitigation measures.

Lines 125-129: Did weight keep constant?

Lines 158-160: Does other factors, such as wind direction, wave height, weight (if constant during this experimental period), influence the sink rate?

Lines 182-183: It was found that reducing setting speed reduced line tension and resulted in gear sinking closer to the vessel (Pierre et al. 2013). Is it the case in this study?

Lines 288-289: Need references.

Lines 312-324: There have been sufficient studies investigating the impacts of sink rate of longline fisheries on seabird bycatch (e.g. Cortés et al. 2018). Methods such as weighted lines, Chilean system, thawing the bait, smaller distance between consecutive weights have been adopted to increase sink rate in order to reduce seabird bycatch, although these methods have been found with some operational problems. For example, an increase of entanglements between the branch lines and hooks may happen during the setting operations when additional weights were used. More discussion on how to increase sink rate and the corresponding impacts on fish catch will improve this paper. A study found that weighted lines increased blackmouth catshark catches, possibly because catsharks forages in the near bottom layer and on the seabed (Anastasopoulou et al. 2013). An introduction on the life history of the target species European Hake would help to discuss the potential impacts of increased sink rate.

Figures 4 and 5: The four panels for each figure can be put in one page, using some R packages such as ggplot2.

References:

Pierre JP, Goad DW, Thompson FN, Abraham ER. Reducing seabird bycatch in bottom-longline fisheries. Final Report prepared for the Department of Conservation: Conservation Services Programme projects MIT2011-03 and MIT2012-01. Department of Conservation, Wellington. 2013.

Anastasopoulou A, Mytilineou C, Lefkaditou E, Dokos J, Smith C., Siapatis P, et al. Diet and feeding strategy of blackmouth catshark Galeus melastomus. J Fish Biol. 2013; 83: 1637–1655.

Cortés V, González-Solís J. Seabird bycatch mitigation trials in artisanal demersal longliners of the Western Mediterranean. PLoS One. 2018;13(5): e0196731.

Reviewer #5: This is an excellent manuscript and was a pleasure to review. I have a few minor suggestions for consideration:

- line 83. Provide a reference for 'Clean Catch UK'

- consider combining Figures 1 and 3

- lines 130-132. Consider rewriting as follows: Fishers were asked to deploy gear as they would in a normal fishing operation. Eight to ten TDRs were deployed on each experimental set, depending on fishing conditions.

- line 139. Delete 'reportedly the most bycaught species in the north east Atlantic floated demersal longline fishery', as this repeats infomation provided earlier.

- line 182. insert 'that' between stern and different, i.e., ..... from the stern that different positions ....

- line 255. Spelling - polygon.

- reference 41. Please provide more details, e.g., name of university.

6. PLOS authors have the option to publish the peer review history of their article (what does this mean?). If published, this will include your full peer review and any attached files.

Reviewer #1: No

Reviewer #2: No

Reviewer #3: No

Reviewer #4: No

Reviewer #5: **Yes: **Barry Baker

---

## [Author Response · Author response to Decision Letter 0]

18 Feb 2022

Thank you for giving us the opportunity to improve greatly our manuscript, based on constructive comments from the reviewers. We believe we have now answered to all the issues which required Major Revisions, in particular in relation to the statistical analysis and in light of the PLOS data policy & Funding information requirements. Response to all specific comments can be found in the submitted File "Response to Reviewers_Rouxel et al."

We hope you will find the revised version of the Manuscript and Figures meet the quality and publication requirements of the PLOS journal.

---

## [Decision Letter · Decision Letter 1]

18 Mar 2022

PONE-D-21-37923R1Slow sink rate in floated-demersal longline and implications for seabird bycatch riskPLOS ONE

Dear Dr. Rouxel,

Thank you for submitting your manuscript to PLOS ONE. After careful consideration, we feel that it has merit but does not fully meet PLOS ONE’s publication criteria as it currently stands. Therefore, we invite you to submit a revised version of the manuscript that addresses the points raised during the review process.

We look forward to receiving your revised manuscript.

Kind regards,

Vitor Hugo Rodrigues Paiva, Ph.D.

Academic Editor

PLOS ONE

Journal Requirements:

Reviewers' comments:

Reviewer's Responses to Questions

**Comments to the Author**

1. If the authors have adequately addressed your comments raised in a previous round of review and you feel that this manuscript is now acceptable for publication, you may indicate that here to bypass the “Comments to the Author” section, enter your conflict of interest statement in the “Confidential to Editor” section, and submit your "Accept" recommendation.

Reviewer #1: All comments have been addressed

Reviewer #2: All comments have been addressed

Reviewer #3: All comments have been addressed

Reviewer #4: All comments have been addressed

2. Is the manuscript technically sound, and do the data support the conclusions?

Reviewer #1: Yes

Reviewer #2: Yes

Reviewer #3: Yes

Reviewer #4: Yes

3. Has the statistical analysis been performed appropriately and rigorously? 

Reviewer #1: Yes

Reviewer #2: Yes

Reviewer #3: Yes

Reviewer #4: Yes

4. Have the authors made all data underlying the findings in their manuscript fully available?

Reviewer #1: Yes

Reviewer #2: Yes

Reviewer #3: Yes

Reviewer #4: Yes

5. Is the manuscript presented in an intelligible fashion and written in standard English?

Reviewer #1: Yes

Reviewer #2: Yes

Reviewer #3: Yes

Reviewer #4: Yes

6. Review Comments to the Author

Reviewer #1: The authors have done a good job revising the manuscript. That some of the environmental variables not showing up as significant as we thought they would be in some of the models may be because a single set of covariates were recorded for each longline set, which includes multiple TDR tracks (see Table 1), and the relatively narrow geographical extent (Fig 2), thus limiting the contrast in the recorded covariates. Extended monitoring involving multiple vessels operating across different seasons may be needed in further work, preferably with concurrent catch and bycatch monitoring. Nonetheless, the result is clear that the sink rate at non-weighted sections is substantially slower than the ACAP recommendation.

I have just some minor editorial comments:

1, Please conform to journal guidelines on tables. "Tables must be editable, cell-based objects" instead of images.

Reviewer #2: (No Response)

Reviewer #3: Thank you for addressing all comments. I only add a couple of minor suggestions:

- line 228 - later in line 384, a specific reference is given in a different format, i.e. Cortés & González-Solís [51]. Please double check journal guidelines.

- line 266 - might be more clear if the specific tables were listed.

Reviewer #4: The manuscript “Slow sink rate in floated-demersal longline and implications for seabird bycatch risk” was resubmitted with extensive revisions. The authors have effectively responded to the various comments and suggestions and the manuscript is much improved. In particular, I found the extensive revisions of the discussion very helpful, as well as improved descriptions of models used. I only have a few minor comments as follows:

Lines 156-157: Merge to the last paragraph. It is weird for a sentence to be a paragraph.

Line 225: There is no need to state the full name of credible intervals, because it has appeared before in Line 197.

Line 234: Change “is” to “was”.

Lines 660-661: Add journal information: New Zealand Journal of Marine and Freshwater Research, 36(1), 185-195, DOI: 10.1080/00288330.2002.9517079.

7. PLOS authors have the option to publish the peer review history of their article (what does this mean?). If published, this will include your full peer review and any attached files.

Reviewer #1: No

Reviewer #2: No

Reviewer #3: No

Reviewer #4: No

---

## [Author Response · Author response to Decision Letter 1]

30 Mar 2022

All reviewers' comments were addressed in the attached "Response to Reviewers_Rouxel et al_2V" document.

---

## [Editor Report · Decision Letter 2]

4 Apr 2022

Slow sink rate in floated-demersal longline and implications for seabird bycatch risk

PONE-D-21-37923R2

Dear Dr. Rouxel,

We’re pleased to inform you that your manuscript has been judged scientifically suitable for publication and will be formally accepted for publication once it meets all outstanding technical requirements.

Kind regards,

Vitor Hugo Rodrigues Paiva, Ph.D.

Academic Editor

PLOS ONE
---

## [Editor Report · Acceptance letter]

11 Apr 2022

PONE-D-21-37923R2 

Slow sink rate in floated-demersal longline and implications for seabird bycatch risk 

Dear Dr. Rouxel:

I'm pleased to inform you that your manuscript has been deemed suitable for publication in PLOS ONE. Congratulations! Your manuscript is now with our production department. 

Kind regards, 

on behalf of

Dr. Vitor Hugo Rodrigues Paiva 

Academic Editor

PLOS ONE